

# Short physical performance battery as a predictor of mortality in community-dwelling older adults: a longitudinal study in the Brazilian Amazon region

Caroline de Fátima Ribeiro Silva, Daniela Gonçalves Ohara, Areolino Pena Matos, Ana Carolina Pereira Nunes Pinto and Maycon Sousa Pegorari

Universidade Federal do Amapá, Macapá, Brazil

## ABSTRACT

**Background**. The Short Physical Performance Battery (SPPB) is an instrument for assessing physical performance widely used in research among the elderly in multiple settings. We did not find Brazilian longitudinal studies that aimed to analyze the predictive capacity and accuracy of the SPPB among community-dwelling older adults and no systematic reviews were found on the accuracy of the SPPB in predicting mortality in community- dwelling older adults. This study aimed to analyze the capacity and accuracy of the SPPB for predicting mortality in community-dwelling older adults, as well as to determine cut-off points for men and women.

**Method**. Longitudinal observational study conducted with 411 (70.1 ± 7.25 years) community-dwelling older adults, between 2017 and 2020 (37.7 ± 6.24 months). Physical performance was evaluated using the SPPB and information on the all-cause mortality rate was also recorded. Multivariate Cox regression analyses and curves were performed using the Kaplan–Meier method. Receiver Operating Characteristic (ROC) curves were constructed, with the parameters of area under the ROC curve (AUC) to determine cutoff points for discriminating mortality, considering a significance level of 5% ($p < 0.05$) and 95% confidence interval (CI) 95%.

**Results**. Older adults with very low and low physical performance in the SPPB, showed higher risks of mortality (HR = 9.67; 95% CI [1.20–77.65]; HR = 4.06; 95% CI [1.09–15.01]), respectively. In the subtest's analysis, older adults with low performance in the balance (HR = 0.54; 95% CI [0.36–0.81]) and gait speed tests (HR = 0.50; 95% CI [0.33–0.76]) showed greater risks of dying. The same was reproduced for categories in each test (participants that scored 2 points in the balance test had an HR = 5.86; 95% CI [1.84–18.61] and 2 points in the gait speed test, HR = 5.07; 95% CI [1.76–14.58]. The cutoff point ≤ 9 in the SPPB set the discriminator criterion for mortality in older people of both sexes.

**Conclusions**. The SPPB, as well as the balance and gait speed subtests were predictors of mortality, and the SPPB is accurate in predicting mortality among community-dwelling older adults.

Corresponding author
Maycon Sousa Pegorari,
mayconpegorari@yahoo.com.br

## INTRODUCTION

Aging leads to lifelong accumulation of cellular damage that results in a gradual decline in physical and mental ability (*Rudnicka et al., 2020*). As a result, older adults are at an increased risk of frailty, functional decline, and other adverse health outcomes, as well as death (*Fried et al., 2001*). As a result, interest is increasing in finding tests that can be used as screening tools for early identification of people who may benefit from targeted interventions. Several studies have suggested that the assessment of physical performance may be useful in the clinical evaluation of older patients, especially because it may lead to early identification of individuals at a heightened risk of adverse health outcomes (*Pahor et al., 2014*; *Pavasini et al., 2016*).

Two previous meta-analyses showed that sit-to-stand, balance time, and gait speed tests separately were able to discriminate community-dwelling older adults at an increased risk of dying (*Cooper et al., 2010*; *Studenski et al., 2011*). Interestingly, another study suggests that combining the three tests may enhance the prognostic value of these components in predicting mortality (*Nofuji et al., 2016*).

Considering these results, some observational studies have consistently found an association between physical performance assessed with the Short Physical Performance Battery (SPPB) and the incidence of disability, institutionalization, and hospital admission (*Freiberger et al., 2012*; *Panas et al., 2013*; *Pahor et al., 2014*; *Patrizio et al., 2021*). The SPPB is an easy-to-apply instrument that includes sit-to-stand, balance time, and gait speed tests and has been used to assess the level of physical performance and functional capacity in older adults in different settings (*Treacy & Hassett, 2018*; *Silva et al., 2021*).

In line with these findings, a systematic review with meta-analysis analyzed the relationship between the SPPB score and all-cause mortality in adults and found an association between low scores and a higher risk of death (*Pavasini et al., 2016*). The study also suggested that a score <10 predicted all-cause mortality in adults (*Pavasini et al., 2016*). Taken together, these findings may support the application of therapeutic strategies, tailoring more intensive interventions to individuals with low physical performance. However, the systematic review evaluated adult individuals and analyzed SPPB scores in categories (0–3, 4–6, 7–9, 10–12) and not as a continuous variable, so the suggested score (<10) may not precisely discriminate older adults at a heightened risk of mortality.

Furthermore, most of the longitudinal studies on the association of SPPB and mortality in older adults were conducted in North America, Europe, and Asia (*Silva et al., 2021*). To the best of our knowledge, only one study was carried out with older adults treated at an outpatient clinic in South America (*Fortes-Filho et al., 2020*). Specifically in Brazil, no longitudinal studies have been conducted aiming at analyzing SPPB as a predictor of mortality among community-dwelling older adults and no systematic reviews were found on the accuracy of the SPPB in predicting mortality in community-dwelling older adults.

Therefore, the present study aimed to analyze the capacity and accuracy of the SPPB for predicting mortality in community-dwelling older adults, as well as to determine cut-off points for men and women.

## MATERIAL AND METHODS

### Analytical observational cohort study

This is a longitudinal observational study carried out using data collected in a previous survey (baseline 2017) that evaluated community-dwelling older adults from Macapa, capital of the State of Amapa, Brazil. Information on sample size calculation and population characteristics are available in a previous study (*Ohara et al., 2018*). This study was approved by the Research Ethics Committee of the Federal University of Amapa under protocols numbers 1.738.671 and 4.444,628. All the participants agreed to participate in the research by signing an In-formed Consent Form.

### Inclusion and exclusion criteria

Elderly individuals aged 60 years or older, able to walk, with or without gait aids, were included in this study. Individuals that could not be located after three attempts, had neurological or orthopedic sequelae, presented cognitive decline, were hospitalized, or had health conditions that prevented physical tests, were excluded. The older adults were recruited and assessed at their respective homes in the year 2017, and interviews were conducted face-to-face by properly trained undergraduate students, and monitored by field supervisors (researcher teachers) (*Silva et al., 2020*). A total of 443 older adults were interviewed, of which 27 had cognitive decline and five did not complete the assessments, and for these reasons were excluded. Thus, this study was based on a sample of 411 older adults (baseline 2017).

In 2020, the participants were contacted at home by telephone or in person. Of these, there were 34 deaths and 41 older adults were not located for the following reasons: change of address/house, they were not contacted at the residence after three visits, and the address/residence was not found, totaling 336 survivors identified (Fig. 1).

### Physical performance (independent variable)

Physical performance was assessed using the Brazilian version of the Short Physical Performance Battery (SPPB), translated to Brazilian Portuguese (*Nakano, 2007*). The SPPB consists of three sequential tests that assess balance (static), strength (lower limbs), and gait. Detailed description of SPPB and cut-off points can be viewed in a previous publication (*Silva et al., 2021*).

*Balance*–The balance test consists of three different positions. (1) standing position with feet together; (2) standing with one foot partially forward; and (3) standing with one foot forward.

*Strength*—evaluated with the sit-to-stand test. The time that the participant took to complete the movement five times was evaluated, and the shorter the time, the better the performance.

*Gait*—assessed using the gait speed test. The time that the participant took to cover a distance of 4 m was registered, in which a shorter time indicated better performance in the test.

Each test is scored from 0 (inability to perform the task) to 4 points (best performance in the test). The total score ranges from 0 (worst performance) to 12 points (best performance)

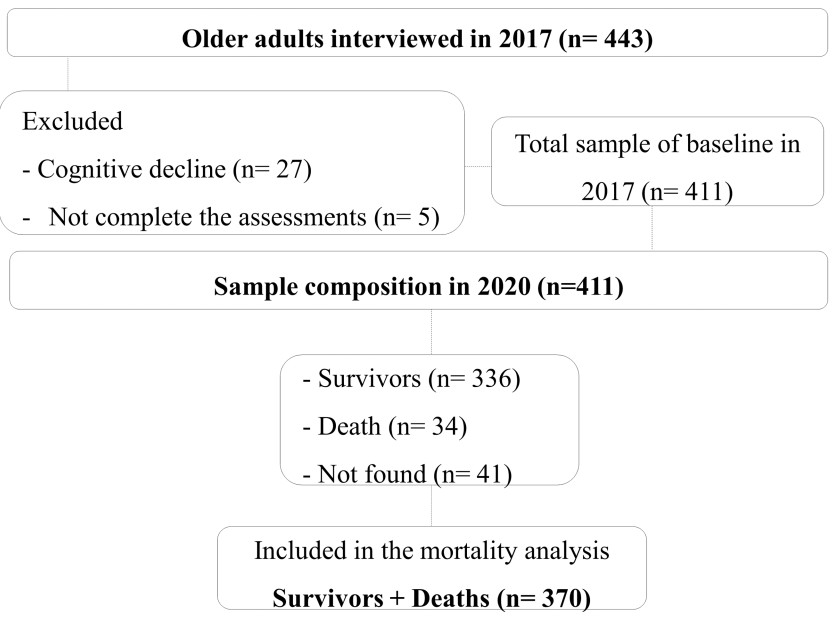

**Figure 1 Sample composition 2017–2020.**

(*Nakano, 2007*). The SPPB was operationalized as a continuous and categorical variable. As a categorical variable four classifications were considered: scores from 0–3 (very low performance), 4–6 (low performance), 7–9 (moderate performance), 10–12 (good performance).

## Mortality (dependent variable)

Data regarding mortality (death) were obtained by consulting the National Register of the Deceased (CNF) at http://www.falecidosnobrasil.org.br. In addition, the older adults and/or family members were contacted via telephone or directly at home to confirm deaths and the number of survivors. Deaths occurring after the baseline assessments in 2017 and in the subsequent three years (2018, 2019, and 2020) were considered. Therefore, a temporal function was adopted: the date of the last assessment and the date of death.

## Adjustment variables

The following were considered as adjustment variables: (i) socioeconomic variables: sex, age (in years), marital status (without a partner or with a partner), education (years), individual income (no income, up to one minimum wage, or two or more minimum wages), and housing arrangement (living alone or accompanied); (ii) health variables: number of diseases, number of medications regularly used, self-perceived health, falls and hospitalizations in the previous year; (iii) functional disability for basic and instrumental activities of daily living through the Katz and Lawton and Brody scales, adapted to the Brazilian reality (*Lino et al., 2008*; *Santos & Virtuoso Junior, 2008*), physical activity level using the International Physical Activity Questionnaire (IPAQ), adapted for older adults from Brazil by *Benedetti, Mazo & Barros (2004)* and *Benedetti et al. (2007)*, depressive

symptoms by the abbreviated Geriatric Depression Scale (GDS-15) (*Almeida & Almeida, 1999*), and the body mass index (BMI: Body Mass/Height$^2$-kg/m$^2$).

## Statistical analysis

Data were analyzed using absolute frequencies and percentages for categorical variables and means and standard deviations for quantitative variables. To compare categories of interest the chi-square test was used. To compare the remaining study variables, the student-t and one-way ANOVA tests were used with Dunnett T3 or Bonferroni correction for multiple comparisons of sociodemographic and health variables between the survivors, deaths, and non-located groups; and between the SPPB categories (very low, low, moderate, and good).

The predictive capacity of the SPPB and its subtests was established through multivariate Cox regression analysis, with death as the outcome. In addition, survival analysis was established using the Kaplan–Meier method, considering a 95% confidence interval and a significance level of 5% ($p < 0.05$), using the Statistical Package for Social Sciences program (SPSS), version 25.0. To determine the cut-off points to discriminate mortality, Receiver Operating Characteristic (ROC) curves were constructed, with the parameters of area under the ROC curve (AUC), using MedCalc software, version 11.4.4.

## RESULTS

Among the 411 older adults evaluated in 2017, 8.3% ($n = 34$) corresponded to deaths and 81.8% ($n = 336$) to survivors during the mean follow-up of 37.7 $\pm$ 6.24 months (2017–2020). The sociodemographic and health characteristics, as well as the scores obtained in the SPPB and in the subtests, according to deaths and survivors, are presented in Table 1.

The causes of death were: (i) complications from diabetes; (ii) unspecified viral respiratory infections and pneumonia; (iii) malignant neoplasms; (iv) cardiovascular disease; (v) cerebrovascular accident–CVA; (vi) Covid-19; and (vii) other unspecified acute clinical conditions.

In the log rank test, it was observed that the older adults with the worst performance in the scale had a significant lower probability of survival compared to the older adults with the best performance levels ($p = 0.002$). Similarly, older adults with lower scores in the balance ($p = 0.009$) and gait speed ($p < 0.001$) subtests had a lower probability of survival compared to older adults with better performance. There was no significant difference for the sit-to-stand test ($p = 0.421$) (Fig. 2).

The adjusted Cox regression model indicated that the total SPPB score remained associated with mortality (HR = 0.72; 95% CI [0.58–0.88]; $p = 0.002$). In an analysis by SPPB categories, older adults with very low (score 0-3) and low (score 4–6) performance had higher risks of death with HR = 9.67 (95% CI [1.20–77.65]; $p = 0.033$) and HR = 4.059 (95% CI [1.09–15.01]; $p = 0.008$) respectively. For the three subtests, the balance (HR = 0.54; 95% CI [0.36–0.81]; $p = 0.003$) and gait speed tests (HR = 0.50; 95% CI [0.33–0.76]; $p = 0.001$) were identified as predictors of death. Regarding the subtest categories (score 0 to 4), the balance (HR = 5.86; 95% CI [1.84–18.61]; $p = 0.003$) and gait speed tests (HR =
**Table 1  Socioeconomic and health characteristics of community-dwelling older adults according to groups of survivors, deaths, and not located, Macapá-AP, Brazil (2017–2020).**

| Variables | Survivors 336 (81.7) | Deaths 34 (8.3) | Not found 41 (10.0) | p | Total n = 411 |
|---|---|---|---|---|---|
| **Age (years)** | 70.14 ± 7.27 | 71 ± 7.33 | 69.4 ± 7.08 | 0.641[b] | 70.1 ± 7.25 |
| **Sex** | | | | | |
| Male | 112 (33.3) | 17 (50.0) | 9 (22.0) | **0.037**[a] | 138 (23.6) |
| Female | 224 (66.7) | 17 (50.0) | 32 (78.0) | | 273 (66.4) |
| **Education (years)** | 5.95 ± 5.30 | 5.17 ± 5.24 | 5.02 ± 4.83 | 0.436[b] | 5.79 ± 5.25 |
| **Living arrangement** | | | | | |
| Living alone | 22 (6.5) | 4 (11.8) | 2 (4.9) | 0.497[a] | 28 (6.8) |
| Accompanied | 314 (93.5) | 30 (88.2 ) | 39 (95.1) | | 383 (93.2) |
| **Income** | | | | | |
| No income | 30 (8.9) | 3 (8.8) | 11 (26.8) | **0.036**[a] | 44 (10.7) |
| Up to one minimum wage | 158 (47.0) | 18 (52.9) | 17 (41.5) | | 193 (47.0) |
| Two or more minimum wages | 148 (44.0) | 13 (38.2) | 13 (31.7) | | 174 (42.3) |
| **Marital status** | | | | | |
| No partner | 156 (46.4) | 19 (55.9) | 16 (39.0) | 0.346[a] | 192 (46.5) |
| With partner | 180 (53.6) | 15 (44.1) | 25 (61.0) | | 220 (53.5) |
| **Number of diseases** | 5.41 ± 2.92 | 5.91 ± 2.93 | 5.19 ± 2.76 | 0.546[b] | 5.43 ± 2.90 |
| **Number of medications** | 1.66 ± 1.79 | 1.50 ± 1.39 | 1.31 ± 1.55 | 0.830[b] | 1.62 ± 1.74 |
| **BMI (kg/m$^2$)** | 27.9 ± 4.72 | 29.4 ± 5.49 | 27.9 ± 6.30 | 0.286[b] | 28.1 ± 4.97 |
| **Depressive symptoms** | | | | | |
| Yes | 57 (17.0) | 12 (35.3) | 10 (24.4) | **0.024**[a] | 79 (19.2) |
| No | 279 (83) | 22 (64.7) | 31 (75.6) | | 332 (80.8) |
| **BADL** | | | | | |
| Dependent | 22 (6.5) | 7 (20.6) | 1 (2.4) | **0.015**[a] | 30 (7.3) |
| Independent | 314 (93.5) | 27 (79.4) | 40 (97.6) | | 381 (92.7) |
| **IADL** | | | | | |
| Dependent | 105 (31.3) | 7 (20.6) | 13 (31.7) | 0.429[a] | 125 (30.4) |
| Independent | 231 (68.8) | 27 (79.4) | 28 (68.3) | | 286 (69.6) |
| **Health perception** | | | | | |
| Positive | 101 (30.1) | 8 (23.5) | 15 (36.6) | 0.470[a] | 124 (30.2) |
| Negative | 234 (69.9) | 26 (76.5) | 26 (63.4) | | 286 (69.8) |
| **Hospitalization in previous 12 months** | | | | | |
| Yes | 46 (13.7) | 6 (17.6) | 6 (14.6) | 0.815[a] | 58 (14.1) |
| No | 290 (86.3) | 28 (82.4) | 35 (85.4) | | 353 (85.9) |
| **Falls in the previous 12 months** | | | | | |
| Yes | 71 (21.1) | 7 (20.6) | 5 (12.2) | 0.404[a] | 83 (20.2) |
| No | 265 (78.9) | 27 (79.4) | 36 (87.8) | | 328 (79.8) |
| **Level of physical activity** | | | | | |
| Inactive | 154 (45.8) | 23 (67.6) | 16 (39.0) | **0.029**[a] | 193 (47.0) |
| Active | 182 (54.2) | 11 (32.4) | 25 (61.0) | | 218 (53.0) |

(*continued on next page*)

**Table 1** (*continued*)

| Variables | Survivors 336 (81.7) | Deaths 34 (8.3) | Not found 41 (10.0) | p | Total n = 411 |
|---|---|---|---|---|---|
| Total SPPB[*] | 9.35 ± 1.88 | 8.14 ± 2.51 | 9.04 ± 2.19 | **0.003**[b] | 9.22 ± 2.00 |
| **SPPB subtests** | | | | | |
| Balance | 3.71 ± 0.66 | 3.35 ± 1.01 | 3.63 ± 0.91 | **0.020**[b] | 3.67 ± 0.73 |
| Gait speed[*] | 3.62 ± 0.70 | 3.14 ± 1.07 | 3.51 ± 0.84 | **0.002**[b] | 3.57 ± 0.76 |
| Sit-to-stand | 2 ± 1.19 | 1.61 ± 1.12 | 1.90 ± 1.22 | 0.177[b] | 1.96 ± 1.19 |

**Notes.**

Outcomes with significant results are shown by *p*-values in bold.

n (%), Mean (standard deviation); BMI, Body mass index; BADL, basic activities of daily living; IADL, instrumental activities of daily living.

[a] Chi square test

[b] One-way ANOVA tests were used with Dunnett T3 or Bonferroni correction for multiple comparisons. Significant differences were observed between the groups:

[*] survivors ≠ deaths ($p < 0.05$).

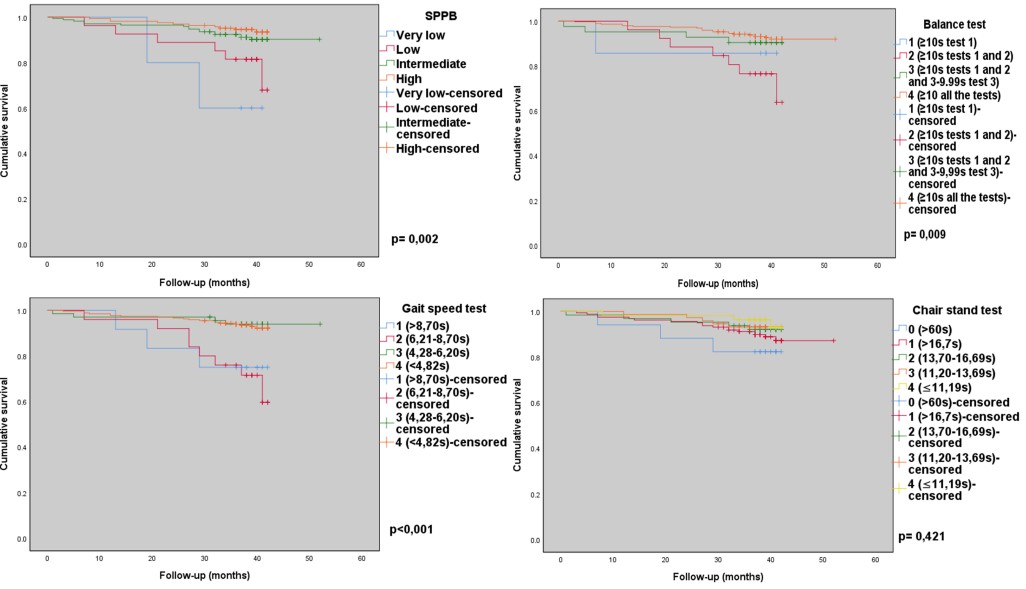

**Figure 2** Survival curves for SPPB categories and subtests among community-dwelling older adults, *n* = 370. Macapa-AP, Brazil (2017–2020).

5.07; 95% CI [1.76–14.58]; $p = 0.003$) remained significantly associated with death even in the adjusted model (Table 2).

Figure 3 demonstrates that for men, the area under the ROC curve was 0.655 (95% CI [0.57–0.74]; $p = 0.044$), with a sensitivity of 70.59% and a specificity of 58.93%. For women, the value of the area under the ROC curve was 0.667 (95% CI [0.60–0.73]; $p = 0.029$), with a sensitivity of 70.59% and specificity of 40.18%. The results of the area under the ROC curve represents a weak discrimination capacity (*Hosmer & Lemeshow, 2000*). The cutoff point ≤9 on the SPPB scale was able to predict mortality in older adults of both sexes.

**Table 2  Cox regression model for the SPPB as a predictor for the risk of mortality among community-dwelling older adults, $n = 370$, Macapa-AP, Brazil (2017–2020).**

| SPPB | Mortality | | | | | |
|---|---|---|---|---|---|---|
| | HR$_{unadjusted}$ | CI 95% | $p$ | HR$_{adjusted}$ | CI 95% | $p$ |
| **SPPB (score)** | 0.78 | 0.68–0.89 | **0.001** | 0.72 | 0.58–0.88 | **0.002** |
| **SPPB (categories)** | | | | | | |
| Very low (0–3) | 8.13 | 1.78–37.2 | **0.007** | 9.67 | 1.20–77.65 | **0.033** |
| Low (4–6) | 3.94 | 1.43–10.8 | **0.008** | 4.05 | 1.09–15.01 | **0.008** |
| Moderate (7–9) | 1.57 | 0.71–3.47 | 0.259 | 1.33 | 0.53–3.30 | 0.536 |
| Good (10–12) | | 1 | | | 1 | |
| **SPPB (subtests)** | | | | | | |
| **Balance (score)** | 0.61 | 0.44–0.85 | **0.004** | 0.54 | 0.36–0.81 | **0.003** |
| **Balance (0–4 points)** | | | | | | |
| 1 (≥10s test 1) | 2.08 | 0.28–15.46 | 0.473 | 2.65 | 0.19–36.13 | 0.464 |
| 2 (≥10s tests 1 and 2) | 3.91 | 1.67–9.17 | **0.002** | 5.86 | 1.84–18.61 | **0.003** |
| 3 (≥10s tests 1 and 2 and 3–9.99s test 3) | 1.30 | 0.44–3.78 | 0.625 | 2.02 | 0.57–7.17 | 0.274 |
| 4 (≥10s all the tests) | | 1 | | | 1 | |
| **Gait speed (score)** | 0.56 | 0.40–0.77 | **<0.001** | 0.50 | 0.33–0.76 | **0.001** |
| **Gait speed (0–4 points)** | | | | | | |
| 1 (>8.70s) | 4.00 | 1.18–13.55 | **0.026** | 4.29 | 0.69–26.37 | 0.116 |
| 2 (6.21-8.70s) | 4.94 | 2.16–11.29 | **<0.001** | 5.07 | 1.76–14.58 | **0.003** |
| 3 (4.82-6.20s) | 0.84 | 0.28–2.47 | 0.754 | 0.52 | 0.14–1.91 | 0.326 |
| 4 (<4.82s) | | 1 | | | 1 | |
| **Sit-to-stand (score)** | 0.75 | 0.54–1.02 | 0.071 | 0.83 | 0.57–1.19 | 0.322 |
| **Sit-to-stand (0–4 points)** | | | | | | |
| 0 (>60s) | 3.52 | 0.71–17.47 | 0.123 | 0.69 | 0.07–6.89 | 0.756 |
| 1 (>16.7s) | 2.13 | 0.62–7.24 | 0.224 | 1.43 | 0.33–6.13 | 0.623 |
| 2 (13.70–16.69s) | 1.50 | 0.36–6.31 | 0.574 | 1.91 | 0.41–8.90 | 0.406 |
| 3 (11.20–13.69s) | 1.26 | 0.30–5.28 | 0.750 | 1.29 | 0.28–5.94 | 0.743 |
| 4 (≤11.19s) | | 1 | | | 1 | |

**Notes.**

Outcomes with significant results are shown by $p$-values in bold.

HR, hazard ratio; 95% CI, confidence interval $p < 0.05$; SPPB, Short Physical Performance Battery; 1, reference category; s, (seconds).

Adjusted for age (years), sex, education (years), income, marital status, living arrangement, number of diseases and medications, hospitalization in the previous year, health perception, functional disability, level of physical activity, falls, body mass index, and depressive symptoms.

## DISCUSSION

The present study analyzed the accuracy of the Short Physical Performance Battery for predicting mortality in a representative sample of community-dwelling older adults at a mean follow-up of three years and two months, establishing sex-specific cutoff points. This study also identified that low physical performance in the SPPB, as well as in the balance and gait speed subtests, were associated with lower survival and a higher risk of death.

It is worth noting that, to our knowledge, this is the first study with Brazilian community-dwelling older adults aimed at addressing the relationship between physical performance assessed by the SPPB and mortality, demonstrating cutoff points for predicting this outcome (*Silva et al., 2021*). Additionally, the sociodemographic characteristics of the older adults in

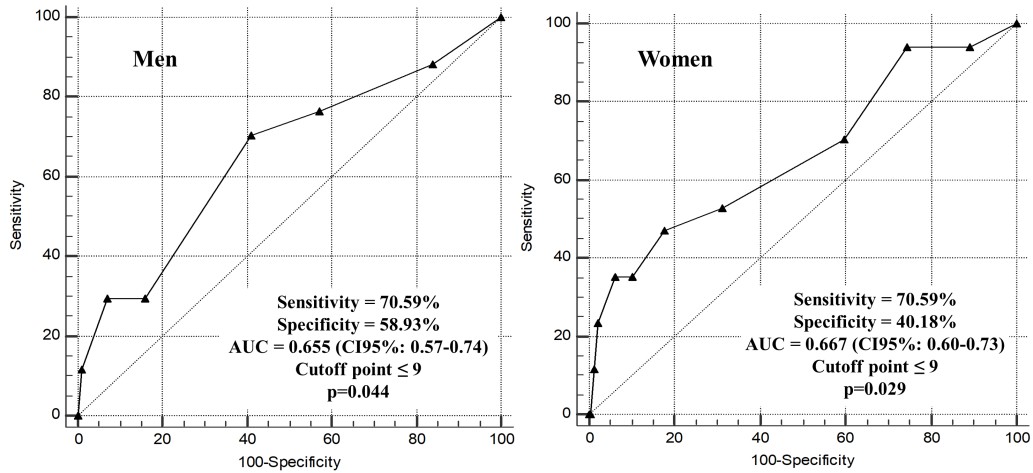

**Figure 3** Areas under the ROC curve for SPPB as a predictor of mortality among community-dwelling older men and women, *n* = 370. Macapa-AP, Brazil (2017–2020).

this study are similar to the investigation conducted with Mexican-American older adults (*Mutambudzi et al., 2019*), which suggests the possibility of comparing and extending the data to other Latin American countries.

We emphasize that data referring to cutoff points (accuracy) by ROC (AUC) curve analyses of the SPPB for mortality among community-dwelling older adults were not reported in any study carried out in the American continent (*Silva et al., 2021*). The cutoff points found in this study could help to identify older adults with a higher risk of death at an early stage, given the easy applicability of the SPPB in different settings.

Poor physical performance among older adults is influenced by multifactorial aspects (chronic, psychological, social, and environmental factors) to which individuals are exposed throughout life, and has been associated with adverse health outcomes such as hospitalization, institutionalization, disability, and death. Thus, physical performance is an essential element for the detection of age-related clinical conditions (*Patrizio et al., 2021*) and the SPPB has shown good prognostic value for these short- and long-term outcomes, even among individuals with moderate performance (*Cesari et al., 2009*; *Arnau et al., 2016*; *Fortes-Filho et al., 2020*). These aspects were reinforced by studies conducted with community-dwelling older adults, with or without previous clinical conditions (*Legrand et al., 2014*; *Brown, Harhay & Harhay, 2015*; *Fox et al., 2015*; *Lattanzio et al., 2015*; *Landi et al., 2016*).

In the current study, the survival curves for mortality indicated that participants in lower categories in the SPPB were associated with a lower probability of survival compared to those in higher categories, and the same was reproduced for the balance and gait speed tests (total score) and in the analysis by subtest categories (maximum score 4 points). Several studies conducted with community-dwelling older adults (*Rolland et al., 2006*; *Cesari et al., 2009*; *Legrand et al., 2014*; *Mutambudzi et al., 2019*), hospitalized (*Chiarantini et al., 2010*; *Lamers et al., 2017*; *Nastasi et al., 2018*; *Van Mourik et al., 2019*; *Saitoh et al., 2020*),

and after discharge from intensive care units (*Corsonello et al., 2012*; *Lattanzio et al., 2015*) are in line with our findings.

Most of these studies were conducted in European countries, such as Italy (*Perera et al., 2006*; *Cesari et al., 2009*; *Lattanzio et al., 2015*; *Landi et al., 2016*; *Veronese et al., 2017*), England (*Fox et al., 2015*), Finland (*Björkman et al., 2019*), and France (*Rolland et al., 2006*), with follow-up times ranging from 24 months to 10 years. The only Brazilian longitudinal study on the SPPB and mortality with older adults was carried out by *Fortes-Filho et al. (2020)* and was conducted with 512 older outpatients with acute illnesses followed for one year. All these studies indicate significant risk values for low or moderate SPPB scores as predictors of mortality.

Not only in the SPPB, but also observed in isolation, the balance, gait speed, and strength are all components strictly-related to physical performance (*Silva et al., 2021*). In young adults, lean muscle mass comprises approximately 50% of total body weight but this drops to around 25% upon entering the age range of 75–80 years (*Short & Nair, 2000*). Age-related muscle loss is paramount in the reduction in physical performance associated with age, as it may lead to decreased muscle strength (*Cruz-Jentoft et al., 2010*). Gait and balance disorders in older adults, in turn, are usually multifactorial. However, loss of muscle strength may be associated with these disorders and, together, may predispose the older adult to a higher risk for adverse outcomes such as fractures from falls, hospitalization, frailty, and death (*Nofuji et al., 2016*).

In surveys of community-dwelling older adults conducted in France (*Rolland et al., 2006*) and in the United States (*Verghese et al., 2012*) in a follow-up of 3.8 years and 32 months, respectively, the gait speed test was more strongly associated with mortality when compared to the total SPPB score. Similarly, other studies suggest that the gait speed test alone is able to predict mortality as well as the SPPB, even among older adults in different settings (*Cesari et al., 2013*; *Pamoukdjian et al., 2017*; *Veronese et al., 2017*). On the other hand, the study by *Fortes-Filho et al. (2020)* showed the SPPB total score to be the best discriminator of adverse outcomes (including death) when compared to the gait speed test alone.

In contrast, *Charles et al. (2020)*, in a study carried out in Belgium with 604 institutionalized older adults, concluded that an increase of 1-unit in the balance test was able to reduce the probability of death by 12% during a 3-year follow-up. Similar results were reported by *Nastasi et al. (2018)* in a study with 142 older adults hospitalized in the United States, in a 5-year follow-up, in which the balance test was more strongly associated with mortality. However, in both studies, no significant results were observed for the other scale tests (gait speed and sit-to-stand tests).

Another study (*Cesari et al., 2008*) included 335 Italians aged 80 years or more, living in the community and followed up for 24 months. In the comparative analysis of the SPPB components, the authors found that the sit-to-stand test showed a greater prognostic value for mortality compared to balance and gait speed. This result contrasts with the findings of the present study, in which the sit-to-stand test was not associated with the risk of death among the older adults evaluated. Above all, it is worth considering that the sit-to-stand test in isolation has been shown to be a predictor of mortality among older adults (*Barbour*

*et al., 2016*; *Keevil et al., 2018*) and a recent systematic review highlighted its predictive value for functional dependence in ADLs in this group of individuals (*Wang et al., 2020*). Of note, it would be useful to establish the predictive power of isolated tests of the SPPB for mortality, considering that there are advantages in terms of time and costs to performing one tests in an isolated manner compared to the entire SPPB (*Rolland et al., 2006*).

The SPPB is a widely used instrument to assess physical performance in scientific research, as it demonstrates a high level of reliability regarding the measurement of physical function among community-dwelling older adults (*Freiberger et al., 2012*; *Treacy & Hassett, 2018*). It is emphasized that small changes in the scale such as 0.5 points already express significant results, even if small; and changes of 1 point reflect a substantial impact on the global functional capacity of the older adults (*Treacy & Hassett, 2018*).

The present study not only identified the discriminating capacity of the SPPB for mortality but also established SPPB cutoff points for predicting this outcome. This study identified a cutoff point in the total SPPB score that is lower ($\leq 9$) and this finding is consistent with reported in previous systematic review (*Pavasini et al., 2016*), indicating that the increased risk of mortality may be identified using different cutoff values in different settings and ages. Of note, one study conducted in Italy with 506 older adults discharged from hospital, followed for 1 year, identified a cutoff point $<5$ in the area under the ROC curve (AUC - 0.66; sensitivity: 0.66 and specificity: 0.62) as a predictor of mortality (*Corsonello et al., 2012*).

This study has some limitations: (i) the SPPB scale was applied only at baseline and participants were not reassessed during the study follow-up, which made it impossible to know about possible changes in the level of physical performance over time; (ii) the use of questionnaires and self-reported measures (clinical and health conditions) may not precisely estimate some of the information found. However, the strengths of this study include the use of a representative sample of community-dwelling older adults from the Brazilian Amazon region and the results obtained provide relevant information about physical performance and mortality in this group of individuals. In addition, the findings demonstrate the ability of the SPPB to predict the risk of death and survival, as well as its accuracy; presenting cutoff points for both sexes. Finally, this investigation also evidences the ability of the balance and gait speed tests to predict survival and the risk of death.

In this perspective, this study can guide clinicians and policy makers in the decision-making process, especially when aiming at the implementation of interventions targeting reductions in adverse health outcomes in the geriatric population in Brazil. On the other hand, it is emphasized that these data should not be used and interpreted indiscriminately, and factors such as the clinical context, the heterogeneity of the decline in physical performance from the analysis of extrinsic and intrinsic aspects acquired throughout life should be considered (*Hoekstra et al., 2020*).

Moreover, it is important to know that the SPPB has advantages over other physical performance assessment instruments, as it is a non-invasive, low-cost, and easily applied instrument that can be used in different settings. However, despite the consistent results of the SPPB for predicting adverse health outcomes, longitudinal studies aiming at establishing

cutoff points in Brazilian older adults are still needed to ratify the findings on the risk of death in this population.

## CONCLUSION

Older people with worse physical performance in the SPPB (total score) and in the categories of very low (scores 0–3) and low performance (scores 4–6) had a greater risk of death compared to those with better performance. The balance and gait speed subtests configured mortality predictors, and the cutoff point ≤9 was demonstrated to be a mortality discriminator for both men and women.

### Funding
This work was supported by the Foundation for Research Support of the State of Amapá (No. 250.203.029/2016). The funders had no role in study design, data collection and analysis, decision to publish, or preparation of the manuscript.

### Grant Disclosures
The following grant information was disclosed by the authors:
Foundation for Research Support of the State of Amapá: 250.203.029/2016.

### Competing Interests
The authors declare there are no competing interests.

### Author Contributions
- Caroline de Fátima Ribeiro Silva performed the experiments, analyzed the data, prepared figures and/or tables, and approved the final draft.
- Daniela Gonçalves Ohara conceived and designed the experiments, performed the experiments, analyzed the data, authored or reviewed drafts of the article, and approved the final draft.
- Areolino Pena Matos analyzed the data, authored or reviewed drafts of the article, and approved the final draft.
- Ana Carolina Pereira Nunes Pinto conceived and designed the experiments, analyzed the data, authored or reviewed drafts of the article, and approved the final draft.
- Maycon Sousa Pegorari conceived and designed the experiments, analyzed the data, authored or reviewed drafts of the article, and approved the final draft.

### Human Ethics
The following information was supplied relating to ethical approvals (i.e., approving body and any reference numbers):

This study was approved by the Research Ethics Committee of the Federal University of Amapa under protocols numbers 1.738.671 and 4.444,628.

## Data Availability

The raw measurements are available in the Supplementary File.

## Supplemental Information

Supplemental information for this article can be found online at http://dx.doi.org/10.7717/peerj.13630#supplemental-information.

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
