# Peer review of "Short physical performance battery as a predictor of mortality in community-dwelling older adults: a longitudinal study in the Brazilian Amazon region"

_PeerJ, doi:10.7717/peerj.13630_

## Round 0.1 · original submission · Minor Revisions

Dear authors, since reviewer 1 recommended accepting your article, reviewer 2 asked for major revisions and reviewer 3 asked for minor revisions, it is my opinion as the academic editor for your article that it requires minor revisions to be accepted.

·

Basic reporting

The article was written using clear and technically correct language. Literature background is sufficient and supports the hypothesis and the research problem. All complementary information and files were well provided, making the study reproducible and reliable. Methods were well described and the results support both objectives and conclusion.

Experimental design

The knowledge gap was presented with enough literature support, which pointed to the relevance of the results for clinical practice in Gerontology and Geriatrics. Methods was appropriatedly described and seem feasible and reliable, in accordance with the best practice in epidemiology research.

Validity of the findings

Findings is novel and relevant for clinical practice in Gerontology and Geriatrics. The SPPB is widely used in research and practice, then the information about its cutoff point and the relationships with mortality increase the interest with regard older adult's physical functioning despite chronic diseases.

Additional comments

Please, check the reference citation in the line 65. There is a typo.

Reviewer 2 ·

Basic reporting

- This is an observational, analytical cohort study with a significant sample of elderly people. The topic is not new, but it contributes with some data to the area of gerontology.
- English is adequate and understandable.
-Tables and figures were presented properly.
- References used are updated.

ABSTRACT
- Line 18- They must not state that studies were not carried out, due to the speed of publications currently. I suggest “We did not find Brazilian longitudinal studies that aimed to analyze the predictive capacity and accuracy of the SPPB in the mortality of the elderly in the community”.
- Line 21- The wording of the objective is a little confusing. I suggest: “This study aimed to analyze the predictive capacity and accuracy of the SPPB in the mortality of the elderly in the community, as well as to determine cut-off points for men and women”.
- Line 22 -Use only the abbreviation SPPB, since its abbreviation has already been defined previously.
- Include in the methodology which level was considered statistically significant or indicative of significance.
- Line 26 - Include in the methodology which level was considered statistically significant or indicative of significance.

INTRODUCTION
- Many studies on the importance of the SPPB test were discussed. I suggest decreasing and increasing the number of solid references that reinforce the contextualization of the theme. Why is the SPPB test an important screening tool for the elderly? As a suggestion, it is possible to add more about the various changes associated with aging, bringing more physiological information about what happens. Even so, the reason for choosing this test should be reinforced, in contrast to other tests, in which environment it has been used (home, hospital, research).
- Line 50 - I suggest citing “Various studies” rather than “Various authors”. And include at least three studies in the reference.
-Line 65 - Correct this reference in the text and in the final references. It is misquoted. You should also include it within the same parentheses as the previous authors.
- Line 76 - Correct the reference as mentioned above.

Experimental design

METHODOLOGY
- Original research, with a defined research question. Well-described statistical analysis.
- The procedures performed were described, but there is no information regarding the training of the researchers responsible for data collection (to ensure the quality of the evaluation protocol), nor if the evaluations were applied on the same day and if the participants practiced a recognition of the test before the main assessment.
- Design and sample: Correct to “Analytical observational cohort study”.
- Inclusion and exclusion criteria: I suggest reversing the wording. "Elderly individuals aged 60 years or older, able to walk, with or without gait aids, were included in this study." It is not necessary to quote the informed consent form again. The wording of Figure 1 is confusing. I suggest improving. Only 370 elderly people were followed from 2017 to 2020?
- Physical performance (independent variable): What was the cut-off point used in the sit-to-stand test and in the gait test?

Validity of the findings

RESULTS
- All underlying data has been provided; are robust, statistically sound and controlled.
- In Table 1 and 2, I suggest placing a margin in the classifications of each variable, it is better to observe.
- On lines 176 and 177: Point out that it was statistically significant even though it has the p-value.
- In the paragraph of Figure 3: The p value was significant for both curves, however the classification of the area under the curve was not presented. I suggest reading “Metz CE. Basic principles of ROC analysis. Semin Nucl Med.1978;8(4):283-98.” and present the result of the area according to the classification proposed by the author.

DISCUSSION
- First paragraph of the discussion: The purpose of the study has already been mentioned. The first paragraph of the discussion should point out the most relevant findings of the study and contribution to the area. Include some references that reinforce these findings.
- Line 201: As already mentioned we shouldn't point this out, due to the speed of publications nowadays. It's a risky statement. If it is the will of the authors, it is possible to indicate that so far there are still few studies. It may be interesting to comment on the strengths of the study as well, such as originality and methodological rigor.
- Line 207: Change "It is worth mentioning" to "We emphasize that", so as not to be repetitive.
- Line 208: As already mentioned we should not point this out, due to the speed of publications nowadays.
- Line 233: We cannot say this, due to the speed of publications currently.
- Line 238: This paragraph became vacant in the middle of the discussion. I suggest linking to the results found in the search.

- Line 247: In this paragraph the same happened as in the introduction. Studies that used the SPPB are brought up. But within the clinical and practical context, what is the importance of the results found? I suggest pointing out that these variables cannot be used indiscriminately, as there are biopsychosocial factors that involve balance, gait and strength, such as individual health characteristics, intrinsic and environmental factors. Bring references and briefly discuss other variables involved.
- Line 255: In this paragraph the same thing happened as in the introduction. Studies that used the SPPB are brought up. Little discussion about the results.
- Excellent closing of the discussion and conclusion of the study.

Additional comments

Overall the article is well written. I highlight the organization of the methodology and presentation of the results. However, the introduction should be improved, as well as the discussion, which should address the importance of the data found and not just the similarity with other studies.

Reviewer 3 ·

Basic reporting

Literature references, sufficient field background/context provided. Only in the Introduction: Lines 75-76 it is necessary to put which studies.
Professional article structure, figures, tables. Raw data shared.
Self-contained with relevant results to hypotheses.

Experimental design

Well-defined, relevant, and meaningful research question. It is stated how the research fills an identified knowledge gap.
Important suggestions in methodology:
It is necessary that the methods need to be written with more details: How were the tests applied? face-to-face or over the phone? It needs to be clearer how the evaluation occurred in 2017.
It is important to make it clear in the methodology that the contact in 2020 was only to know the deaths, at the beginning of the methodology (lines 104-108 are a little confusing). In this same paragraph, there is no need to put the results of deaths in the methodology, because they already are in the results. It is clearer to put lines 104-108 together with the item project and sample, before the inclusion and exclusion criteria.
Line 117: It is Important to describe how many repetitions are necessary to do in the sit-to-stand test.

Validity of the findings

Results: appropriated.
All underlying data has been provided; they are robust, statistically sound, and controlled.
Discussion: Meaningful replication encouraged where rationale and benefit to literature is clearly stated.
It is necessary to put references in lines: 201-203; and 238.
Conclusion: The conclusion must be properly formulated. It must be related to the original question investigated, and must be limited to those supported by the results. Is it important to further argue a little more the conclusion. About how it can be used, or how important the study was. Needs for further studies? What does this study bring positive? Make an overview of the main results.

Additional comments

I congratulate the authors on tackling this extremely important topic. It is extremely important that we increasingly have tests for the clinical evaluation of the elderly, especially because it can lead to the early identification of individuals with increased risk for various health situations.

---

## Round 0.2 · accepted · Accept

I congratulate the authors on tackling this extremely important topic. I consider the article accepted.

Reviewer 2 ·

Basic reporting

No comments.

Experimental design

No comments.

Validity of the findings

No comments.

Additional comments

Authors responded to requests.

Reviewer 3 ·

Basic reporting

Clear and unambiguous, professional English used throughout.
Literature references, sufficient field background/context provided
Professional article structure, figures, tables. Raw data shared
The requested topics have been reviewed by the authors

Experimental design

Original primary research within Aims and Scope of the journal.
Methods described with sufficient detail & information to replicate.
The requested topics have been reviewed by the authors

Validity of the findings

The requested topics have been reviewed by the authors

Additional comments

I congratulate the authors on tackling this extremely important topic. I consider the article accepted.
It is extremely important that we increasingly have tests for the clinical evaluation of the elderly, especially because it can lead to the early identification of individuals with increased risk for various health situations.